# Granulation of Nickel–Aluminum–Zirconium Complex Hydroxide Using Colloidal Silica for Adsorption of Chromium(VI) Ions from the Liquid Phase

**DOI:** 10.3390/molecules27082392

**Published:** 2022-04-07

**Authors:** Ayako Tabuchi, Fumihiko Ogata, Yugo Uematsu, Megumu Toda, Masashi Otani, Chalermpong Saenjum, Takehiro Nakamura, Naohito Kawasaki

**Affiliations:** 1Faculty of Pharmacy, Kindai University, 3-4-1 Kowakae, Higashi-Osaka 577-8502, Osaka, Japan; 2133420006v@kindai.ac.jp (A.T.); ogata@phar.kindai.ac.jp (F.O.); y.u.pharmafocus@gmail.com (Y.U.); nakamura@phar.kindai.ac.jp (T.N.); 2Kansai Catalyst Co. Ltd., 1-3-13, Kashiwagi-cho, Sakai-ku, Sakai 590-0837, Osaka, Japan; megumu.toda@kansyoku.co.jp (M.T.); masashi.ootani@kansyoku.co.jp (M.O.); 3Faculty of Pharmacy, Chiang Mai University, Suthep Road, Muang District, Chiang Mai 50200, Thailand; chalermpong.saenjum@gmail.com; 4Center of Excellence for Innovation in Analytical Science and Technology for Biodiversity-Based Economic and Society (I-ANALY-S-T_B.BES-CMU), Chiang Mai University, Chiang Mai 50200, Thailand; 5Antiaging Center, Kindai University, 3-4-1 Kowakae, Higashi-Osaka 577-8502, Osaka, Japan

**Keywords:** nickel–aluminum–zirconium complex hydroxide, colloidal silica, granulation, chromium(VI) ion, adsorption

## Abstract

We combined a nickel–aluminum–zirconium complex hydroxide (NAZ) with colloidal silica as a binder to prepare a granulated agent for adsorbing heavy metals from aqueous media. Three samples with different particle diameters were prepared to evaluate the effects on the properties: small (NAZ-S), medium (NAZ-M), and large (NAZ-L). We confirmed the granulation of the prepared samples at a binder content of 25%. NAZ-S had the largest specific surface area and number of hydroxyl groups, followed by NAZ-M and then NAZ-L. Regarding the adsorption capacity, NAZ-S adsorbed the most chromium(VI) ions followed by NAZ-M and then NAZ-L. The binding energy of Cr(2p) at 575–577 eV was detected after adsorption, and the effects of the temperature, contact time, and pH on the adsorption of chromium(VI) ions were evaluated. We identified the following adsorption mechanism: ion exchange with sulfate ions in the interlayer region of the NAZ samples. Finally, the chromium(VI) ions adsorbed by the NAZ samples were easily desorbed using a desorption solution. The results showed that NAZ offers great potential for the removal of chromium(VI) ions from aqueous solutions.

## 1. Introduction

The rapid development of many industries, such as electroplating, metallurgical processes, and mining, has led to increasing concern over the environment and public health, particularly with regard to heavy metals [1]. Among heavy metals, chromium(VI) has been identified as a human carcinogen by international cancer research institutions [2]. Chromium(VI) is considered more toxic than chromium(III), and long-term exposure to or inhalation of chromium(VI) can induce dermatitis, eczema, sneezing, runny noses, nosebleeds, ulcers, kidney damage, and liver damage [3]. Chromium(VI) usually exists in association with oxygen as chromate (CrO_4_^2−^) or dichromate (Cr_2_O_7_^2−^) oxyanions in aquatic environments [4,5]. Therefore, the World Health Organization and United States Environmental Protection Agency set the permissible limits of chromium(VI) to 0.05–0.25 mg/L in drinking water [6], surface water, and industrial wastewater [7]. In Japan, however, chromium is stockpiled along with other heavy metals (e.g., nickel, tungsten, cobalt, molybdenum, manganese, and vanadium) because of its relative rarity and various applications. The United Nations has adopted the 2030 Agenda for Sustainable Development, which suggests the need to recycle rare materials such as metals worldwide. However, the usefulness of recycling rare metals including chromium has not yet been significantly evaluated. The development of techniques for recycling or adsorbing chromium(VI) ions from the liquid phase is important for establishing a sustainable society, including in Japan.

Available treatments for removing chromium(VI) ions from the liquid phase include ion exchange, chemical precipitation, adsorption, reverse osmosis, and biological reduction [8]. The adsorption technique is simple to operate, significantly effective, and economically viable for various heavy metals, including chromium(VI) ions. This technique can also help address sludge problems [9,10].

In addition, various adsorbents such as electrospun DTPA-modified chitosan/polyethylene nanofibers [11], a xanthate-modified magnetic Fe_3_O_4_@SiO_2_-based polyvinyl alcohol/chitosan composite [12], and a chitosan-based adsorbent [13] were reported for the sustainable removal of heavy metals from water and wastewater. These adsorbents have been considered as promising water purification agents for the removal of heavy metals. Additionally, these agents provide various advantages in terms of their biodegradability, biocompatibility, high reactivity, hydrophilicity, and non-toxicity [14]. On the other hand, metal complex hydroxides, which are often called metal layered double hydroxides (LDHs), are useful agents for removing and recovering rare metals from the liquid phase [15,16,17]. We previously reported on the preparation of iron–manganese, nickel–aluminum, and nickel–aluminum–zirconium hydroxides and their adsorption of chromium(VI) ions from the liquid phase [18,19]. The nickel–aluminum complex hydroxide showed efficient removal and recovery of chromium(VI) ions from aqueous media, but it is not suitable for field applications because its powdered form can cause serious problems, including pore clogging, high pressure drops, and mass loss during operation [20]. A practical solution is to combine these metal complex hydroxides with a granulation method for easy and cheap adsorption of chromium(VI) ions. However, the granulation of nickel–aluminum-zirconium complex hydroxides via combination with a binder and the characteristics and adsorption of chromium(VI) ions have not been reported yet.

Therefore, the aims of this study were to combine nickel–aluminum-zirconium complex hydroxides with a binder to develop a granulated water purification agent and assess its potential for adsorbing chromium(VI) ions from the liquid phase.

## 2. Materials and Methods

### 2.1. Materials and Chemicals

A nickel–aluminum–zirconium complex hydroxide (NAZ) was prepared at the molar ratio of Ni^2+^/Al^3+^/Zr^4+^ = 0.9:1.0:0.09. We reported the synthesis process in a previous study [21]. Additionally, sulfate ions were included in the interlayer of NAZ for exchangeable anions. For granulation, colloidal silica (SiO_2_ in H_2_O, Nissan Chemical Co., Tokyo, Japan) was mixed with NAZ and dried at room temperature. Samples were then prepared as shown in Figure 1 by dividing the dried mixture according to particle diameter: NAZ-S (<500 μm), NAZ-M (500–1700 μm), and NAZ-L (>1700 μm). A standard solution of chromium(VI) ions (K_2_Cr_2_O_7_ in 0.1 mol/L HNO_3_) was purchased from FUJIFILM Wako Pure Chemical Co., Osaka, Japan.

Scanning electron microscopy (SEM) and X-ray diffraction (XRD) were performed by using the SU1510 (a tungsten-filament SEM equipped with a secondary electron (SE) detector and a backscatter-electron (BSE) detector, Hitachi Ltd., Tokyo, Japan) and Mini Flex II (Benchtop X-ray diffractometer with advanced detector, Rigaku, Tokyo, Japan) instruments, respectively. The specific surface area was determined by using the NOVA4200*e* instrument (Determined by Brunauer–Emmett–Teller method, Yuasa Ionics, Osaka, Japan). Finally, the pH at the point of zero charge (pH_pzc_) and number of surface hydroxyl groups were measured by using Faria et al.’s method [22] and the fluoride ion adsorption method [23].

To determine the removal mechanism of chromium(VI) ions, the elemental distribution and binding energy were analyzed by using the JXA-8530F (An EPMA featuring a field emission electron gun, JEOL, Tokyo, Japan) and AXIS-NOVA (X-ray photoelectron spectroscopy, Shimadzu Co., Ltd., Kyoto, Japan) instruments, respectively. The numbers of chloride, nitrate, and sulfate ions were measured by using a DIONEX ICS-900 instrument (Thermo Fisher Scientific Inc., Tokyo, Japan). We reported the measurement conditions in a previous study [24]. The anions were prepared using sodium chloride, sodium nitrate, and sodium sulfate, respectively (Guaranteed Reagent, FUJIFILM Wako Pure Chemical Co., Osaka, Japan).

### 2.2. Chromium(VI) Ion Adsorption According to Particle Diameter

The effect of the particle diameter on the adsorption of the chromium(VI) ions was assessed by reacting 0.05 g of each NAZ sample with 50 mL of the chromium(VI) ion solution at a concentration of 50 mg/L for 24 h at an agitation speed of 100 rpm and adsorption temperature of 25 °C. After adsorption, the sample solution was filtrated by using a 0.45-μm membrane filter. The equilibrium concentration of the chromium(VI) ions after adsorption was measured using the iCAP-7600 Duo instrument (Thermo Fisher Scientific Inc., Tokyo, Japan). The adsorption quantity was calculated from the concentrations of chromium(VI) ions before and after adsorption. The data were recorded as the means ± standard error.

### 2.3. Chromium(VI) Ion Adsorption under Different Conditions

The adsorption isotherms were determined by reacting 0.05 g of each NAZ sample with 50 mL of the chromium ion solution at different concentrations (10, 20, 30, 40, and 50 mg/L) for 24 h at an agitation speed of 100 rpm and adsorption temperature of 5, 25, or 45 °C. The effect of contact time on adsorption was assessed by reacting 0.05 g of each NAZ sample with 50 mL of the chromium ion solution at a concentration of 50 mg/L for 1–48 h at an agitation speed of 100 rpm and adsorption temperature of 25 °C. The effect of pH on adsorption was evaluated by reacting 0.05 g of each NAZ sample with 50 mL of the chromium ion solution at a concentration of 50 mg/L for 24 h at an agitation speed of 100 rpm and adsorption temperature of 25 °C. The solution pH was adjusted to 3, 5, 7, 9, and 11 by using either a nitric acid or sodium hydroxide solution. After adsorption, the sample solution was filtrated by using a 0.45-μm membrane filter. The adsorption amount was calculated as previously described in Section 2.2. The data were recorded as the means ± standard error.

### 2.4. Chromium(VI) Ion Adsorption in a Complex Solution System

To clarify the selectivity, 0.05 g of the prepared adsorbent was added to 50 mL of a complex solution including chromium(VI), chloride, nitrate, and sulfate ions, each at a concentration of 1 mmol/L. The reaction solution was shaken for 24 h at an agitation speed of 100 rpm and adsorption temperature of 25 °C before being filtrated using a 0.45-μm membrane filter. The adsorption amount was calculated as previously described in Section 2.2. The data were recorded as the means ± standard error.

### 2.5. Recovery of Chromium(VI) Ions by Desorption

The adsorption/desorption experiment was conducted by reacting 0.1 g of the prepared adsorbent and 50 mL of a chromium(VI) ion solution at a concentration of 100 mg/L. The sample solution was shaken at 100 rpm for 24 h at an adsorption temperature of 25 °C. The adsorption amount of chromium(VI) ions was calculated as previously described in Section 2.2. The NAZ adsorbent was collected after adsorption and then dried at 25 °C for 24 h. The collected NAZ was then reacted with 50 mL of sodium hydroxide solution or sodium sulfate solution at concentrations of 1, 10, or 100 mmol/L. The sample solution was then shaken at 100 rpm for 24 h at a desorption temperature of 25 °C. The sample solution was then filtrated by a 0.45-μm membrane filter. The desorption amount of chromium(VI) ions was calculated from the difference between the initial concentration and equilibrium concentration of chromium(VI) ions.

## 3. Results and Discussion

### 3.1. Characteristics of Prepared Samples

Figure 2 shows SEM images of the prepared samples. NAZ-S had the smallest particle diameter, followed by NAZ-M and then NAZ-L. Thus, we could prepare adsorbents with different particle diameters. However, some of NAZ-S comprised the original NAZ, which indicated that the granulation method with the binder was not perfectly controlled when the particle diameter was too small. A preliminary experiment was performed to determine the optimal binder content for the granulation process. Binder contents of 18%, 25%, and 36% were considered, and granulation of NAZ with 18% binder was difficult because the binder did not mix uniformly. Granulation of NAZ with 36% binder was also very difficult, and we could not modify the particle diameter. Thus, we determined the optimal binder content to be 25%.

Figure 3 shows the XRD patterns of the samples. NAZ and NAZ-S were confirmed to have peaks at (003), (006), (015), and (113), indicating that these samples kept the stacking of the brucite-like sheets. The peaks at (003) and (006) disappeared in NAZ-M and NAZ-L, suggesting that part of the NAZ structure was destroyed by the granulation with the binder (i.e., colloidal silica). The peaks at (015) and (113) simultaneously decreased after granulation. 

Table 1 lists the pH_pzc_, specific surface area, and number of hydroxyl groups of the prepared samples. NAZ-S had a greater specific surface area (101.6 m^2^/g) and number of hydroxyl groups (1.09 mmol/g) than the other NAZ samples, indicating that the granulation with colloidal silica affected the physical properties of the NAZ samples. The pH_pzc_ values showed no obvious differences between samples. Overall, our granulation method allowed us to prepare NAZ samples with different characteristics.

### 3.2. Adsorption Properties

Figure 4 shows the adsorption of chromium(VI) ions by the NAZ samples. NAZ-L adsorbed the least amount of chromium(VI) ions (17.5 mg/g), followed by NAZ-M (25.6 mg/g) and then NAZ-S (27.6 mg/g). We previously reported that NAZ powder (before granulation) adsorbed 24.1 mg/g of chromium(VI) ions under the same experimental conditions [18]. Therefore, the granulation of NAZ with colloidal silica did not influence the adsorption capacity in this study.

We evaluated the relationship between the adsorption amount and adsorbent characteristics, and we confirmed a positive correlation with a correlation coefficient of 0.991. Similar phenomena were observed with the NAZ powder. These results suggest that the adsorption mechanism remained the same before and after granulation. Khitous et al. [25] reported that metal complex hydroxides have various mechanisms for removing harmful heavy metals, such as the interactions between metals and the adsorbent external surface and ion exchange between the target metal and exchangeable anions in the interlayer region. We previously confirmed the interaction between chromium(VI) ions and the external surface of NAZ-S, especially based on the number of hydroxyl groups. Therefore, we determined the binding energies of the NAZ-S surface before and after adsorption, as shown in Figure 5. We successfully detected the Cr(2p) peak at 575–577 eV after adsorption, which supports the above results. Table 2 compares the chromium(VI) ion adsorption capacity of the prepared adsorbents with that of other adsorbents reported in the literature [11,26,27,28,29,30,31]. The prepared NAZ samples had a superior adsorption capacity of chromium(VI) ions from aqueous media than that of other reported adsorbents, excluding ZnNiCr-LDHs, Ni/Fe LDH, and aluminum–magnesium mixed hydroxide. These show that the prepared NAZ samples are potentially applicable to the adsorption of chromium(VI) ions. Additionally, the release of base metals such as Ni, Al, and Zr from the NAZ granulated samples is one of the most important issues in adsorption treatment. This phenomenon directly and strongly affects the adsorption capacity of chromium(VI) ions. Therefore, we evaluated the quantity of base metals released from the NAZ granulated sample in this study. As a result, the quantities of Al and Ni released from NAZ (approximately 0.9 and 8.1 × 10^2^ μg/g for Al and Ni) were greater than that from NAZ-S (not detected and approximately 7.5 × 10^2^ mg/g for Al and Ni), which suggests that NAZ granulated sample was more environmentally friendly (Zr was not detected). This result is very important for the application of NAZ-S in the field.

### 3.3. Effects of Parameters on Chromium(VI) Ion Adsorption

Figure 6 shows the chromium(VI) ion adsorption isotherms of the NAZ samples. Adsorption isotherms show the relationship between the adsorbate and adsorbent concentrations after equilibrium is reached, which can be used to determine the adsorption capacity. The adsorption amount increased significantly with increasing temperature from 5 to 45 °C. These trends are mainly controlled by the activation energy for adsorption. The activation energy of adsorbate increases with increasing temperature [32]. NAZ-L had the lowest adsorption capacity, followed by NAZ-M and then NAZ-S. Additionally, the amounts of chromium(VI) ion adsorbed using NAZ-S were not significantly different between 25 and 45 °C, which suggests that the saturated adsorption capacity was reached under our experimental conditions. The optimal temperature for adsorption was 45 °C, as shown in Figure 6. However, the adsorption capacity slightly increased from 25 to 45 °C. Therefore, further studies are necessary to elucidate the optimal temperature for the adsorption of chromium(VI) ions in detail.

We evaluated the adsorption isotherms using two models: the Langmuir and Freundlich models. The Langmuir model indicates that adsorption occurs uniformly on active sites of the adsorbent, and atoms or ions form a monolayer [33,34]. The Freundlich model indicates that a fundamental empirical equation is fitted to non-ideal adsorption phenomena [35]. The Langmuir and Freundlich models are respectively described below:*q* = *W_s_aC_e_*/(1 + *aC_e_*),(1)
(2)$$\log q=\frac{1}{n}\log C_{e}+\log k,$$
where *q* is the adsorption capacity (mg/g), *W**_s_* is the maximum adsorption quantity (mg/g), and *C_e_* is the equilibrium concentration (mg/L). The adsorption intensity and strength of adsorption are *k* and 1/*n*, respectively. Additionally, *a* is the Langmuir isotherm constant (i.e., binding energy) (L/mg).

Table 3 lists the Langmuir and Freundlich constants for the adsorption of chromium(VI) ions. Both models fitted well to the results (correlation coefficients of >0.940), which indicated that the adsorption mechanism under our experimental conditions was related to a monolayer molecular covering and chemical adsorption [36]. The maximum adsorption amount (*W_s_*) increased with decreasing particle diameter (NAZ-L < NAZ-M < NAZ-S). These trends are similar to those observed for the adsorption isotherms in Figure 6. Finally, the Freundlich constant 1/*n* increased from 0.42 to 0.65 with increasing particle diameter, which indicated that the NAZ samples are suitable for the adsorption of chromium(VI) ions [37].

Figure 7 shows the relationship between the adsorption amount of chromium(VI) ions and the released amount of sulfate ions under isothermal conditions. We confirmed a positive correlation coefficient values of 0.887–0.973, indicating that ion exchange occurred between chromium(VI) ions and sulfate ions in the interlayer region of the NAZ samples. A similar adsorption phenomenon was already reported in our previous work [14]. In particular, some chromate or dichromate oxyanions were exchanged with sulfate ions in the interlayer of the NAZ granulated samples in this study. Additionally, the ion exchange ratios between chromium(VI) ions and sulfate ions using NAZ-S, NAZ-M, and NAZ-L were 0.85, 0.80, and 0.64, respectively (from Figure 7, the slope of linear lines). However, the slope of the linear lines was not 1, indicating that there was some adsorption of chromium(VI) using the NAZ granulated sample in this study (adsorption mechanism was not controlled only by ion exchange).

Figure 8 shows the kinetics of the chromium(VI) ion adsorption for the NAZ samples. The reaction almost reached equilibrium after approximately 24 h under the experimental conditions. After equilibrium at 24 h, no further adsorption was confirmed owing to the lack of vacant sites. Wu et al. [38] previously reported that the time to reach equilibrium is short if the adsorption process is controlled by a single factor.

The experimental kinetic data were simulated using two different kinetic models: the pseudo-first-order and pseudo-second-order models. The linear forms of these models are respectively listed as follows.
ln(*q*_*e,exp*_ − *q*_*t*_) = ln*q*_*e,cal*_ − *k*_1_*t*,(3)
(4)$$\frac{t}{qt}=\frac{t}{qe,cal}+\frac{1}{k2\times qe,cal2},$$
where *q_e,exp_* and *q_e,cal_* are the experimental and calculated adsorption amounts (mg/g), respectively, and *k*_1_ and *k*_2_ are the pseudo-first-order (h^−1^) and pseudo-second-order (g/mg^/^h) rate constants, respectively. Table 4 lists the kinetic parameters for the adsorption of chromium(VI) ions. The correlation coefficient of the pseudo-second-order model (*r* = 0.978–1.000) was higher than that of the pseudo-first-order model (*r* = 0.838–0.989). Furthermore, the theoretical equilibrium adsorption capacity calculated by the pseudo-second-order model agreed with the experimental results, excluding NAZ-L. These phenomena indicate that the rate-limiting step of the adsorption process was controlled by the chemisorption process [39,40,41].

Next, to understand the rate control steps, we adopted the intra-particle diffusion model and evaluated the adsorption process.
*q_t_* = *k_i_**t*^1/2^ + C,(5)
where *k_i_* is the intra-particle diffusion rate constant (mg/g h^1/2^) that can be determined from the straight line plots of *q_t_* against *t*^1/2^. *C* is the intercept of stage *i* associated with the thickness of the boundary layer [42,43]. 

Figure 9 shows the intra-particle diffusion model for the adsorption of chromium(VI) ions. The fitted line does not pass through the original coordinates. In addition, the diffusion mechanism of the adsorption system could be two stages [36], which indicates that the adsorption of chromium(VI) involved more than one kinetic models. The first stage within 4.5 h could be attributed to adsorption over the external surface of the NAZ adsorbent, which was driven by the high initial concentration. In the second stage over 4.4 h, the chromium(VI) ion adsorption rates gradually declined until reaching equilibrium in this study. Additionally, the larger *C* values obtained resulted from the boundary thickness, reflecting the boundary effect [44]. The values of *C*_1_ were smaller compared to *C*_2_, indicating that intra-particle diffusion in the adsorption process was almost at the rate-controlling stage (Table 5). It could be inferred that chromium(VI) ions were adsorbed quickly through electrostatic attraction, ion exchange, and hydroxyl complexation. Similar trends were reported in previous studies [45,46]. Additionally, the *k*_1_ value in the intra-particle model was greater than *k*_2_, indicating that the intra-particle diffusion rate of the first stage was increased owing to the higher adsorption ability of chromium(VI) ions.

Figure 10 shows the effect of pH on the adsorption of chromium(VI) ions. The pH of the solution significantly influences the interactions between heavy metals and adsorbents. Under our experimental conditions, the optimal pH range for the removal of chromium(VI) ions was approximately 5–7. The solution contained different species such as H_2_CrO_4_, HCrO_4_^−^, CrO_4_^2−^, Cr_2_O_7_^2−^, and HCr_2_O_7_^−^ [47]. The adsorption capacity was much lower under acidic conditions (pH 3.0), which may be attributed to the structure being partly destroyed through acidic hydrolysis. In addition, the prepared samples had pH_pzc_ values of 6.2–6.9. Because the pH values were less than 7.0, the sample surfaces were positively charged. This caused the chromium(VI) ions (i.e., oxyoanions) to statically react with the adsorbent surface, and ion exchange between the chromium(VI) and sulfate ions easily occurred in the interlayer region of the NAZ samples. When the pH was 7.0, the surface charge of the NAZ samples became negative. In addition, the concentration of hydroxyl ions increased with the pH. This led to electrostatic repulsion between the chromium(VI) ions and adsorbent surface, and the hydroxyl ions competed with the negatively charged chromium(VI) species for the active sites of the adsorbed surface. Overall, the chromium(VI) ion adsorption mechanism of the NAZ samples can be attributed to electrostatic attraction or ion exchange [28,48].

### 3.4. Chromium(VI) Ion Adsorption Mechanisms

Figure 11 shows the schematic mechanism of chromium(VI) ion adsorption. The adsorption mechanisms can mainly be divided into four stages in this study. First, chromium(VI) ions are captured by the NAZ granulated sample surface due to the surface area or surface charge (physisorption). Second, ion exchange occurs between chromium(VI) ions and sulfate ions in the interlayer of the NAZ granulated sample (chemisorption). Third, electrostatic attraction occurs between surface hydroxyl groups and chromium(VI) ions (chemisorption). Finally, chromium(VI) ions such as oxyanions form (Lewis base) directly bonds to the base metals, such as Ni, Al, and Ze (Lewis acids, indicating surface inner-sphere complex formation (chemisorption). However, further studies are necessary to elucidate the adsorption mechanism of chromium(VI) ions in detail using NAZ samples.

### 3.5. Chromium(VI) Ion Adsorption in the Complex Solution System

Figure 12 shows the selectivity of the NAZ samples regarding chromium(VI) ion adsorption. NAZ-S, NAZ-M, and NAZ-L adsorbed 17.2, 16.7, and 13.2 mg/g, respectively, of chromium(VI) ions. Thus, the adsorption capacity of the NAZ samples decreased by approximately 25–37% with increasing particle diameter. In addition, NAZ-S, NAZ-M, and NAZ-L adsorbed 6.6, 1.8, and 1.7 mg/g, respectively, of chloride ions. Nitrate and sulfate ions were not adsorbed in the experiments. These phenomena suggest that anions such as chloride, nitrate, and sulfate ions did not affect the adsorption capacity of the NAZ samples for chromium(IV) ions in this study. Thus, the NAZ samples offers great potential for the removal of chromium(VI) ions from aqueous solutions.

### 3.6. Adsorption–Desorption of Chromium(VI) Ions in the Desorption Solution

Figure 13 shows the adsorption–desorption using the NAZ samples of chromium(VI) ions with the desorption solution. The adsorbed chromium(VI) ions were easily desorbed using the sodium hydroxide solution or sodium sulfate solution. Moreover, the desorption amount increased with the concentration of the desorption solution from 1 to 100 mmol/L. In particular, the desorption percentage of chromium(VI) ions was >83% with 100 mmol/L sodium hydroxide solution and >66% with 100 mmol/L sodium sulfate solution. Finally, chromium is usually included in stainless steel. Therefore, the recycling process could also be used to prepare chromium for melting or refining processes using an electric furnace.

## 4. Conclusions

In this work, we prepared granulated adsorbents by combining NAZ with colloidal silica. The prepared NAZ samples were then applied to the adsorption of chromium(VI) ions. NAZ-L had the lowest adsorption capacity, followed by NAZ-M and then NAZ-S. The adsorption amount increased with the adsorption temperature and contact time. The optimal pH range for the removal of chromium(VI) ions was approximately 5–7. The obtained data were fitted to the Freundlich and Langmuir models (correlation coefficients: >0.940) and pseudo-second-order model (correlation coefficient: >0.978). Our data showed that the adsorption mechanism is related to ion exchange between chromium(VI) and sulfate ions in the interlayer region of the NAZ samples. The adsorption capacity slightly decreased in the complex solution system. The adsorbed chromium(VI) ions were easily desorbed using a sodium hydroxide solution or sodium sulfate solution, indicating that they can be recovered for rare metal applications. These observations demonstrate the potential of the prepared NAZ samples as agents for the adsorption–desorption of chromium(VI) ions from aqueous solutions.

## Figures and Tables

**Figure 1 molecules-27-02392-f001:**
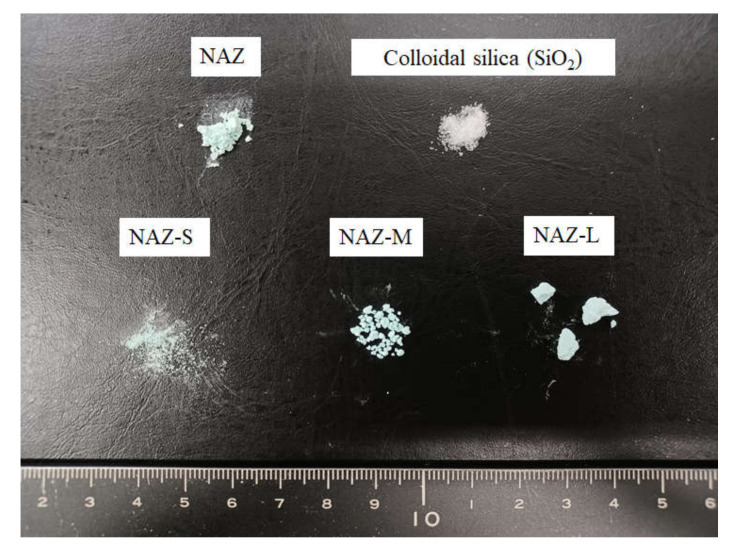
Photograph of prepared adsorbents.

**Figure 2 molecules-27-02392-f002:**
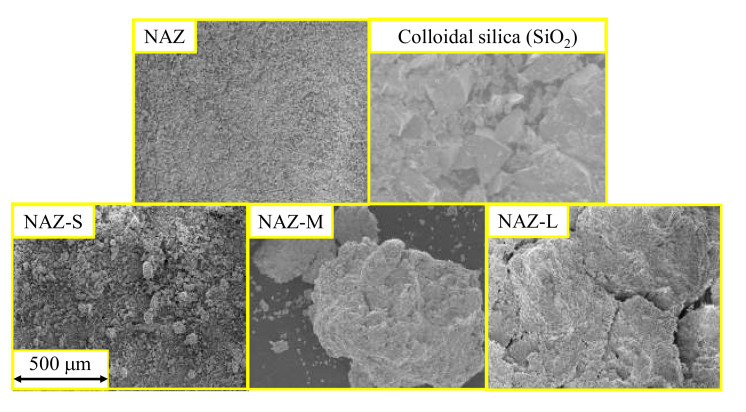
SEM images of prepared adsorbents.

**Figure 3 molecules-27-02392-f003:**
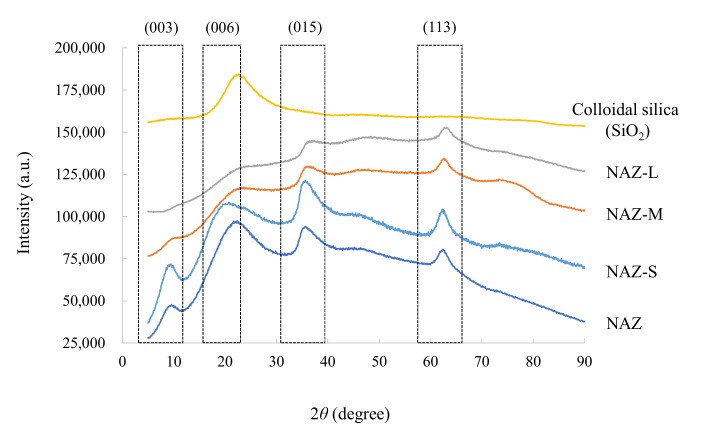
XRD patterns of prepared adsorbents.

**Figure 4 molecules-27-02392-f004:**
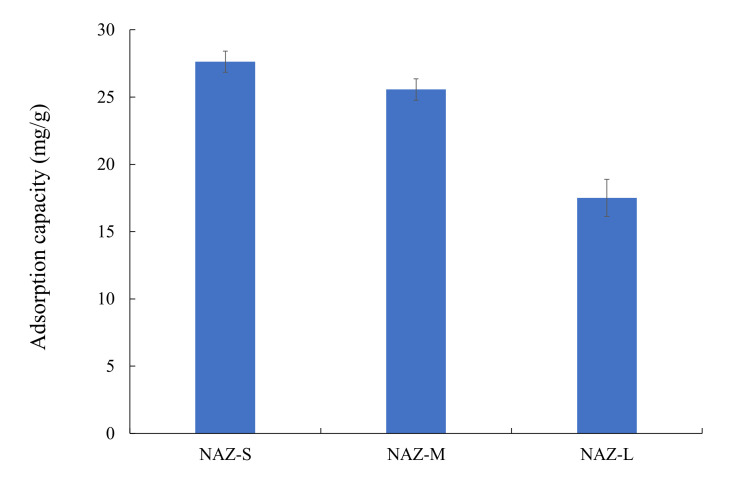
Adsorption of chromium(VI) ions by prepared NAZ-s. Initial concentration: 50 mg/L; sample volume: 50 mL; adsorbent: 0.05 g; temperature: 25 °C; contact time: 24 h; agitation speed: 100 rpm.

**Figure 5 molecules-27-02392-f005:**
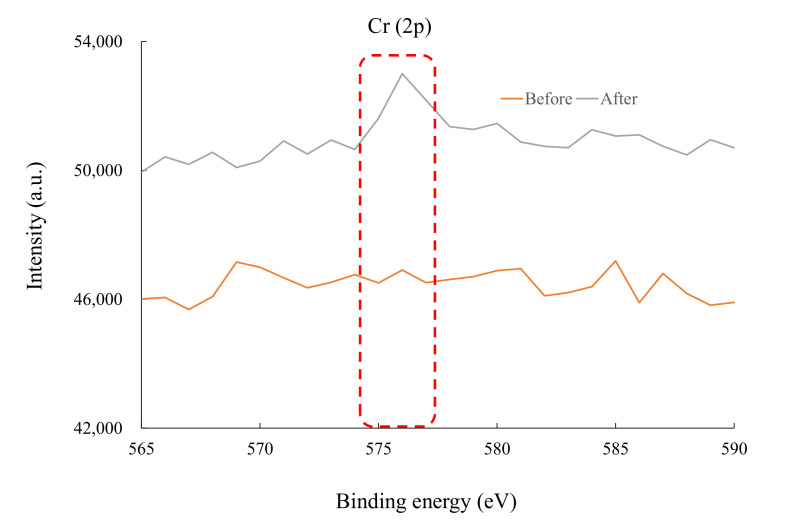
X-ray photoelectron spectroscopy analysis before and after adsorption of chromium(VI) ions.

**Figure 6 molecules-27-02392-f006:**
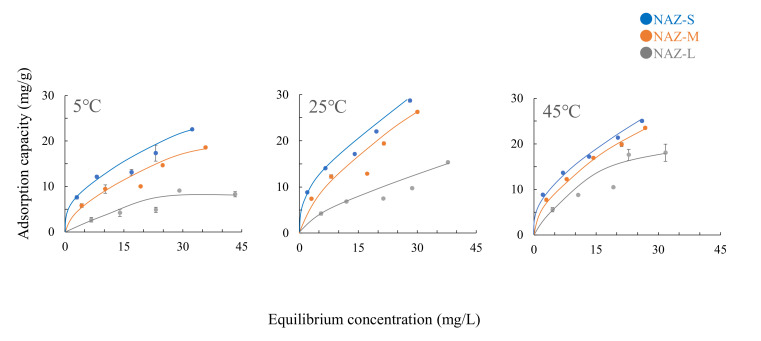
Adsorption isotherms of chromium(VI) ions at different temperatures. Initial concentration: 10–50 mg/L; sample volume: 50 mL; adsorbent: 0.05 g; temperatures: 5 °C, 25 °C, and 45 °C; contact time: 24 h; agitation speed: 100 rpm.

**Figure 7 molecules-27-02392-f007:**
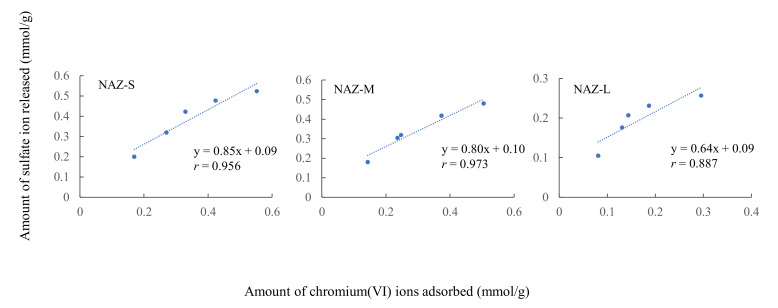
Relationship between adsorption amount of chromium(VI) ions and released amount of sulfate ions.

**Figure 8 molecules-27-02392-f008:**
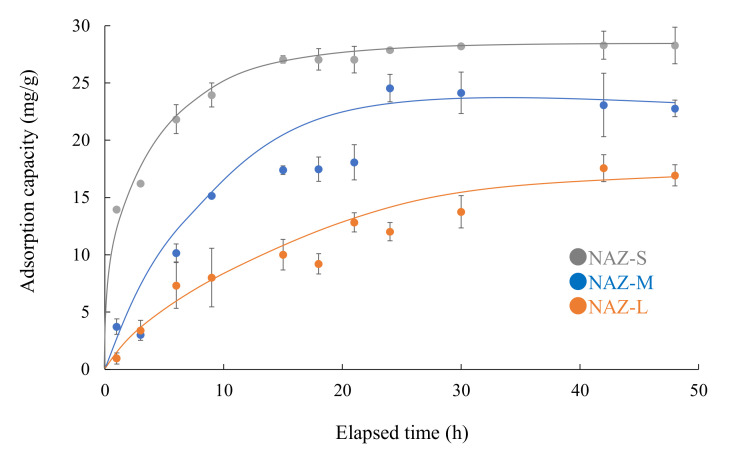
Adsorption of chromium(VI) ions over time. Initial concentration: 50 mg/L; sample volume: 50 mL; adsorbent: 0.05 g; temperature: 25 °C; contact times: 1–48 h; agitation speed: 100 rpm.

**Figure 9 molecules-27-02392-f009:**
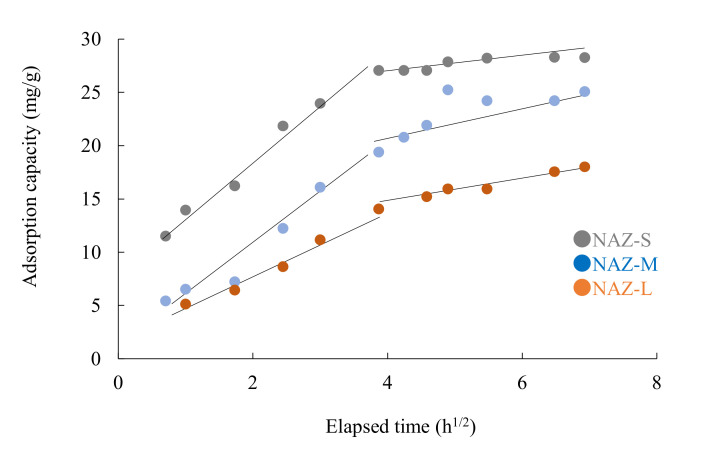
Intra-particle diffusion model for the adsorption of chromium(VI) ions. Initial concentration: 50 mg/L; sample volume: 50 mL; adsorbent: 0.05 g; temperature: 25 °C; contact time: 1–48 h; agitation speed: 100 rpm.

**Figure 10 molecules-27-02392-f010:**
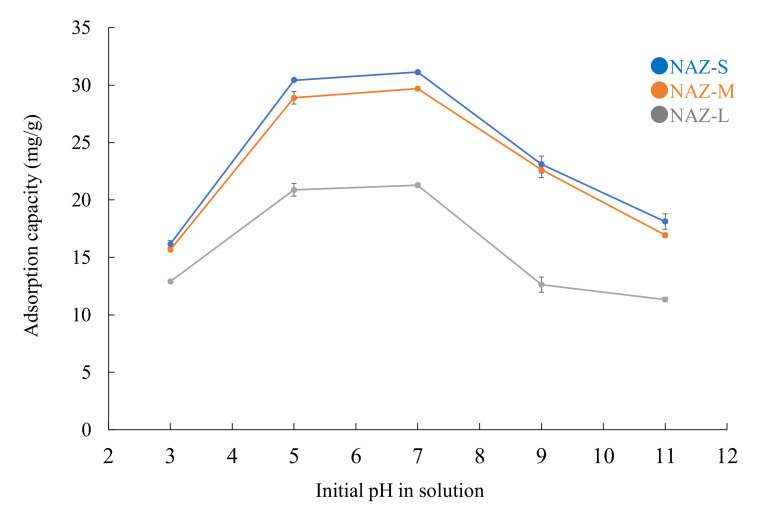
Adsorption of chromium(VI) ions according to pH. Initial concentration: 50 mg/L; sample volume: 50 mL; adsorbent: 0.05 g; temperature: 25 °C; pH: 3, 5, 7, 9, and 11; agitation speed: 100 rpm.

**Figure 11 molecules-27-02392-f011:**
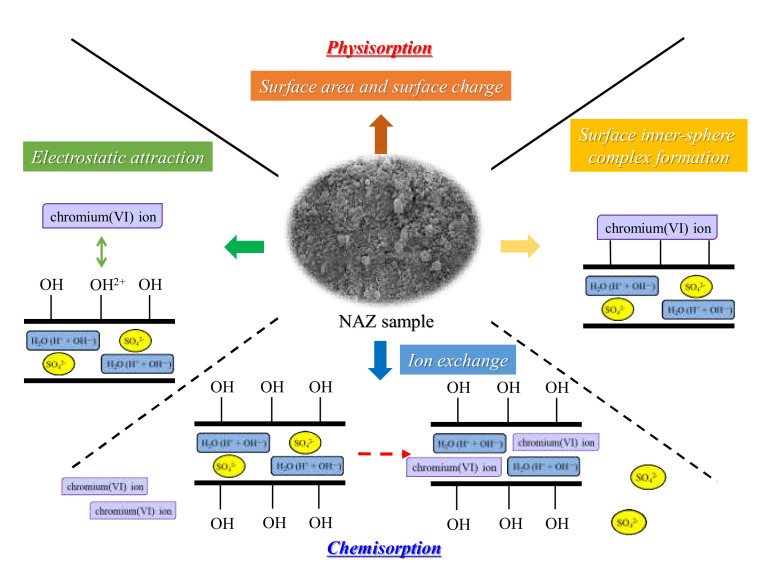
The schematic mechanism of chromium(VI) ion adsorption using NAZ samples.

**Figure 12 molecules-27-02392-f012:**
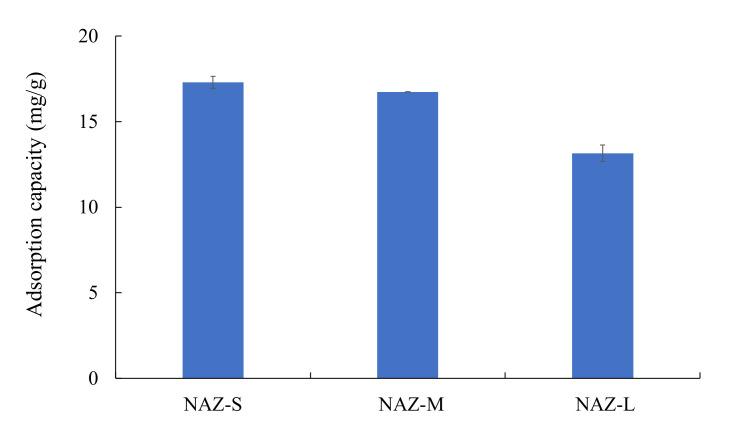
Adsorption of chromium(VI) ions in a complex solution system. Initial concentration: 1 mmol/L; sample volume: 50 mL; adsorbent: 0.05 g; temperature: 25 °C; contact time: 24 h; agitation speed: 100 rpm.

**Figure 13 molecules-27-02392-f013:**
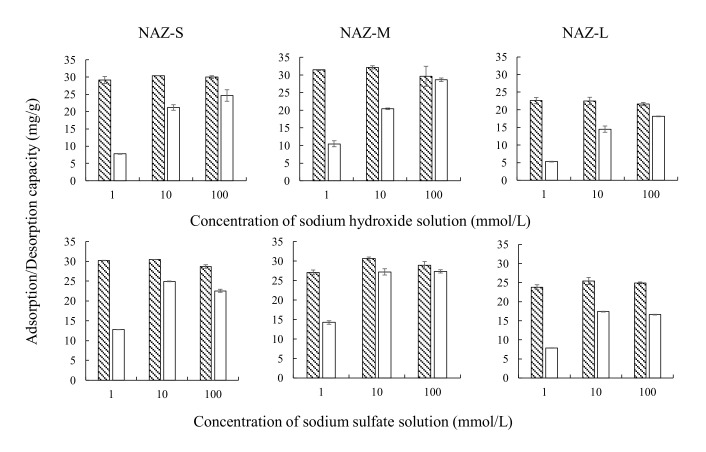
Adsorption–desorption of chromium(VI) ions with a sodium hydroxide solution or sodium sulfate solution. Adsorption conditions: initial concentration: 100 mg/L; sample volume: 50 mL; adsorbent: 0.1 g; temperature: 25 °C; contact time: 24 h; agitation speed: 100 rpm. Desorption conditions: initial concentrations: 1, 10, and 100 mmol/L; sample volume: 50 mL; temperature: 25 °C; contact time: 24 h; agitation speed: 100 rpm. Note: 
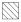
: adsorption; □: desorption; adsorbent: NAZ-S, NAZ-M, and NAZ-L.

**Table 1 molecules-27-02392-t001:** Properties of prepared adsorbents.

Adsorbents	NAZ	Colloidal Silica	NAZ-S	NAZ-M	NAZ-L
pH_pzc_	6.2	5.6	6.9	6.8	6.7
Specific surface area (m^2^/g)	23.8	199.7	101.6	69.5	66.8
Number of hydroxyl groups (mmol/g)	1.08	0.18	1.09	1.08	0.94

**Table 2 molecules-27-02392-t002:** Comparison adsorption capacity of chromium(VI) ions with other reported adsorbents.

Adsorbents	Adsorption Capacity (mg/g)	pH	Temp. (°C)	Initial Concentration (mg/L)	Contact Time(h)	Adsorbent (g/L)	Ref.
Micro-sized granular ferric hydroxide	5.8	7	20	1.2	24	0.04–0.5	[26]
Granular ferric hydroxide	17.0	7	20	20	1	0.2	[27]
ZnNiCr-LDHs	28.2	2	25	30	0.83	0.5	[11]
Ni/Fe LDH	50.43	5	23–55	50	24	0.5	[28]
Aluminum-magnesium mixed hydroxide	105.3–112	2.5–5	30	200	1.5	2	[29]
Maghemite nanoparticle	19.2	2.5	22.5	50	0.25	5	[30]
Zn/Al/Ala LDH	12.25	5.2	25	2	0.25	0.4	[31]
NAZ-S	27.6	7	25	50	24	1.0	This study
NAZ-M	25.6	7	25	50	24	1.0	This study
NAZ-L	17.5	7	25	50	24	1.0	This study

**Table 3 molecules-27-02392-t003:** Freundlich and Langmuir constants for the adsorption of chromium(VI) ions.

Adsorbents	Temperatures (°C)	Langmuir Constants			Freundlich Constants		
		*W_s_* (mg/g)	*a* (L/mg)	*r*	log*k*	1/*n*	*r*
NAZ-S	5	20.75	5.33	0.964	0.67	0.42	0.968
	25	24.81	3.69	0.965	0.81	0.42	0.983
	45	23.92	3.78	0.974	0.80	0.41	0.995
NAZ-M	5	18.87	9.82	0.962	0.43	0.51	0.964
	25	23.20	6.59	0.946	0.62	0.49	0.940
	45	26.32	7.41	0.989	0.64	0.50	0.999
NAZ-L	5	12.53	25.21	0.963	−0.12	0.65	0.938
	25	15.43	14.96	0.962	0.17	0.59	0.949
	45	22.32	13.91	0.962	0.33	0.61	0.954

**Table 4 molecules-27-02392-t004:** Fitting of kinetic data to the pseudo-first-order and pseudo-second-order models.

Adsorbent	*q_e,exp_*	Pseudo-First-Order Model			Pseudo-Second-Order Model		
		*k*_1_ (h^−1^)	*q_e,cal_* (mg/g)	*r*	*k*_2_ (g/mg/h)	*q_e,cal_* (mg/g)	*r*
NAZ-S	28.3	0.14	15.0	0.980	0.02	29.5	1.000
NAZ-M	24.6	0.07	19.4	0.838	3.1 × 10^−3^	27.5	0.984
NAZ-L	18.0	0.06	18.8	0.968	2.1 × 10^−3^	24.0	0.978

**Table 5 molecules-27-02392-t005:** Parameters of the intra-particle diffusion model for chromium (VI) ion adsorption.

Samples	*k*_1_ (mg/g h^1/2^)	*C* _1_	*r*	*k*_2_ (mg/g h^1/2^)	*C* _1_	*r*
NAZ-S	5.0	5.8	0.991	0.5	25.2	0.858
NAZ-M	4.6	1.3	0.980	1.2	17.2	0.696
NAZ-L	3.2	1.3	0.991	1.6	10.0	0.979

## Data Availability

Not applicable.

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
