# Peer review of "Granulation of Nickel–Aluminum–Zirconium Complex Hydroxide Using Colloidal Silica for Adsorption of Chromium(VI) Ions from the Liquid Phase"

_molecules, 2022, doi:10.3390/molecules27082392_

Round 1
Reviewer 1 Report
The paper entitled ” Granulation of nickel–aluminium–zirconium complex hydroxide using colloidal silica for adsorption of chromium(VI) ions from the liquid phase” treats the topic of heavy metal ion adsorption for the treating of waste waters.
The authors prove the adequacy of the granulation method to prepare NAZ adsorbents of different grain sizes which can be used for adsorbing chromium (VI).
The Introduction is adequate, contains the precise objective of the work and the needed introduction to the topic.
Materials and methods
- in 2.1. (and later as well) for most of the named instruments the type of the instrument is not mentioned. For example: what type of instrument is the NOVA4200e? The specific surface area can be determined by more than one methods…
- in 2.1. – lines 84-85: The optimal colloidal silica content is determined as 25%. However later (in 3.1) an explanation is given, the value of 25% puzzles the reader. I suggest to give here a cross reference to the 3.1.
- in 2.3. – line 117 is written: “… different concentrations for 24 h …”. Which are these concentrations?
- in 2.4. subchapter the “selectivity” is mentioned, however the adsorption experiments were not performed in the presence of various metal ions. Therefore the use of the term “selectivity” can be misleading. In some cases the coexisting metal ions can dramatically interfere the adsorption of targeted metals. What can the authors tell about adsorption in presence of other metal ions than Cr(VI)?
- Figure 1. – scale is missing, a darker background would have been better for better visibility
Results and discussions
- Figure 2. – the scale is hardly readable
- line 168: it is obvious that the colloidal silica structure differs from that of NAZ, therefore it doesn’t need to be mentioned
- on the presented figures he authors could consider the use of “Adsorption capacity” instead of “Amount adsorbed”
- Table 2. – instead of adsorption capability is more correct to use adsorption capacity
- line 229-230 – please correct the sentence
- subchapter 3.3., page 9: “q” and “k” are both defined as adsorption capacity, which would question the correctness of the Freundlich equation. Please make difference between the q – equilibrium adsorption capacity (mg/g) and the k – equilibrium Cr(VI) concentration (mg/L)
- Figure 8 – has no legend
- line 310: please correct “adsorption by/desorption”
- line 316: The authors state that the results indicate that Cr(VI) ions adsorbed by NAZ samples can be recycled. What about the reusability of the NAZ adsorbent? How many times can it be reused and how the efficiency of the adsorption is decreasing with the increasing cycle numbers?
- Figure 11: it is not clear what the black columns represent.
The authors prove in this work the adequacy of the NAZ granulated samples to absorb Cr(VI) ions from liquid phase, however the question remains whether this adsorbent is also adequate to be used “in the field”. Does it fulfill the conditions that were listed in the Introduction (lines 66-68)?
The advantages (and eventual disadvantages) of this product are not listed either. If we compare its effectiveness to ZnNiCr-LDHs or Ni/Fe LDH or Al-Mg mixed hydroxides, the question arises: why to use NAZ instead of the highly effective products? Are there environmental considerations to be taken into account?
Reviewer 2 Report
- The English of the text should be checked
- The authors must be included new, relevant and more information about other adsorbent materials (e.g. natural polymer, membranes, resins, montmorillonite). Also, must be included more advantages and disadvantage of adsorption in comparison with other natural polymer. The following references can be included in the Introduction part to improve the quality of manuscript, because they provide relevant information:
https://doi.org/10.1016/j.arabjc.2021.103543; https://doi.org/10.3390/polym14061107; https://doi.org/10.1016/j.seppur.2020.116914
- 1 should be improved, also, explain the purpose of showing Fig. 1. Fig. 1 is not mentioned in the article?
- Lines 45-47; authors write: “Therefore, the World Health Organization and United States Environmental Protection Agency set the permissible limits of chromium(VI) to 0.05–0.25 46 mg/L in drinking water, surface water, and industrial wastewater [6,7].” Please clarify whether it is the World Health Organization or the United States Environmental Protection Agency, also please cite the original source of the data.
- Why NAZ-L had a greater specific surface area (68.4 m2/g) than NAZ-M?
- A schematic mechanism describing the adsorption process must be indicated and included (reactions, interactions etc.)
- The adsorption temperature is 5, 25, and 45°C, why? In addition, it is necessary to compare the adsorption performance of materials at different temperatures, which temperature is the most suitable and why?
- Kinetic data need to add intra-particle diffusion model.
- In Fig. 11 the symbols a, b and c must be cited in the figure caption.
Reviewer 3 Report
Molecules-1651878
In this manuscript, nickel–aluminum–zirconium complex hydroxide (NAZ) with colloidal silica samples with different particle diameters were prepared to evaluate the effects on their properties for adsorbing heavy metal Cr(VI) from aqueous media.
Different adsorption and kinetics experiments were conducted to investigate the adsorption capacity and mechanism of Cr(Vi) binding on available functional groups of NAZ. Further, study of selectivity of adsorbing Cr(Vi) to the NAZ adsorbent under complex environmental conditions and desorption study of Cr(Vi) to the NAZ adsorbent have been studied. This manuscript is well written and provides useful information to the related applications; however, some major issues exist, and the discussion should be improved. I suggest that this manuscript could be considered for publication after significant major revisions are carefully made. The following comments should be useful for authors in revising their manuscript.
Q1) Line 73, it is mention that “the aims of this study were to combine nickel–aluminum complex hydroxides with ….” Is this Ni-Al complex or Ni-Al-Zr complex? Please check same in line 70
Q2) All figures, and tables should move before the information appeared in text
Q3) Line 101 Figure 1 resolution is not enough, improve that and combine the individual images to one image
Q4) Line 118,121 etc, keep the space between number and unit for 25 oC, do apply same throughout the manuscript
Q5) Line 124, It was mention that The solution pH was adjusted to 3, 5, 7, 9, and 11 by using either a nitric acid or sodium hydroxide solution for adsorption studies, Also in line 143…. So I am wondering about two things
- Will NaOH dissolve colloidal silica
- Cr (vi) will precipitate as hydroxide above pH=6 and may contributed to adsorption capacity, this will introduce some overestimation adsorption capacity of Cr(Vi)
Q6) Line 129-131, complex solution system, why author only study Cr(vi) adsorption into NAZ in the presence of Cl-, NO3-, SO42-, how about selectivity study of Cr in the presence of any other heavy metals?
Q7) Line 146-147, author state that “The desorption amount of chromium(VI) ions was calculated from the difference in concentrations of chromium(VI) ions before and after desorption” What does it mean by before desorption ? how you calculate that is unclear
Q8) Figure 2 combine the images, remove scale and other information show in those images, you could use solid yellow line to include the scale, also organize the figure in perfect way
Q9) line 164, What author mean by peaks at (003) (006) etc?
Q10) It is not clear or not explain in the text that why SSA change in following pattern shown in Table 1, NAZ-S> NAZ-L>NAZ-M, how particle size of those samples effect to such change
Q11) XRD spectrum shown in Fig 2d is not scientifically explained although authors state “he successful preparation of LMM was also found through comparative analysis” in Line 144-145. Thus, recommended to improve the text.
Q12) Figure 5, draw Y axis in left, reorganize the Figure 6, 7
Q13) Line 246, it is stated that “chromium(VI) ions and released amount of sulfate ions under isothermal conditions” Although positive correlation is observed, author did not explain how Cr(vi) and SO42- are exchanged, since charge balance and size is not matching, explain this behavior using chemistry into account
Q14) Line 219, Why adsorption capacity increased with increasing temperature (5, 25, 40 oC) for NAZ-M and L but not for NAZ-S shown in table 3.
Effect of temperature on chromium adsorption on NAZ was not properly explained by taking scientific facts into account. Authors should take the effect of Temperature on diffusion etc to explain the observed behavior
Q19) Discuss nonlinear Langmuir equation, apply regression line for chromium adsorption on NAZ (shown in Figure 6). After curve fitting obtain the values you got maximum adsorption and Langmuir constant compare your values with linear regression values, Nonlinear curve fitting introduce less error compared that with linear regression
Q20) In discussion, authors are suggested to include about what can be done after recovering chromium during recycle process
Q21) Authors are encourage to draw some possible interactions (chelating) between Cr(Vi) and NAZ by taking available and suggested functional groups of NAZ into account. Give attention to HSAB theory as well.
Q22) It was not clear chromium adsorption on NAZ is mainly based on physisorption (microporous) or chemisorption (chemical interaction between chromium and adsorbents) or both, so Author should explain by taking both things into account, viewing their structure, functional groups, comparing size of the heavy metal-Cr , and pore width of adsorbent NAZ etc
Refer following Journals
1 DOI https://biomedres.us/fulltexts/BJSTR.MS.ID.002152.php
2 DOI https://doi.org/10.3390/jcs4020057
3 DOI https://doi.org/10.3390/jcs5020046

Round 2
Reviewer 2 Report
accept
Reviewer 3 Report
Authors have carefully address all the comments raised. Thus, I recommended to accept this manuscript in its current form.